# Experiment and Modelling of the Pre-Strain Effect on the Creep Behaviour of P/M Ni-Based Superalloy FGH96

**DOI:** 10.3390/ma16103874

**Published:** 2023-05-21

**Authors:** Hao Wang, Jingyu Zhang, Huashan Shang, Aixue Sha, Yangyang Cheng, Huiling Duan

**Affiliations:** 1Beijing Institute of Aeronautical Materials, AECC, Beijing 100095, China; 2State Key Laboratory for Turbulence and Complex Systems, Department of Mechanics and Engineering Science, BIC-ESAT, College of Engineering, Peking University, Beijing 100871, China; 3HEDPS, CAPT and IFSA Collaborative Innovation Center of MoE, Peking University, Beijing 100871, China

**Keywords:** P/M FGH96 superalloy, creep, pre-strain, micro-twinning

## Abstract

FGH96 is a powder metallurgy Ni-based superalloy used for turbine disks of aero-engines. In the present study, room-temperature pre-tension experiments with various plastic strain were conducted for the P/M FGH96 alloy, and subsequent creep tests were conducted under the test conditions of 700 °C and 690 MPa. The microstructures of the pre-strained specimens after room-temperature pre-strain and after 70 h creep were investigated. A steady-state creep rate model was proposed, considering the micro-twinning mechanism and pre-strain effects. Progressive increases in steady-state creep rate and creep stain within 70 h were found with increasing amounts of pre-strain. Room-temperature pre-tension within 6.04% plastic strain had no obvious influence on the morphology and distribution of γ′ precipitates, although the dislocation density continuously increased with the increase in pre-strains. The increase in the density of mobile dislocations introduced by pre-strain was the main reason for the increase in creep rate. The predicted steady-state creep rates showed good agreement with the experiment data; the creep model proposed in this study could capture the pre-strain effect.

## 1. Introduction

Powder metallurgy (P/M) Ni-based superalloys have been extensively used as turbine discs in modern aeroplane engines due to their excellent mechanical properties. Due to the high requirements of reliability and durability of turbine discs, creep properties of P/M superalloy are of great importance to the safety of the component [1,2,3,4]. Cold pre-deformation is usually used in engineering practice to control residual stress, to improve dimensional stability, and to improve the low cycle fatigue life of aero-engine discs [5,6]. However, pre-deformation also affects the creep properties of Ni-based superalloys at the same time, which might reduce the creep resistance of the disc material [7,8,9]. Therefore, the influence of pre-deformation on the subsequent high-temperature creep behaviour is a topic of significant theoretical and practical interest.

The influence of pre-deformation on creep properties varies for different metallic materials. The improvements in creep resistance after pre-straining were observed for some polycrystalline Ni-based alloys. For example, Marlin et al. [10] found that the creep rate at 760 °C may decrease with increasing pre-strain at the same temperature for an oxide-dispersion-strengthened Ni-based alloy. Cairney et al. [11] reported that pre-strain at both room temperature and elevated temperatures can decrease the creep strain at 520 °C for single-crystal Ni3Al. However, a contrary effect was also observed in Ni-based superalloys where the minimum creep rates were increased and the creep life was shortened with increasing amounts of pre-strain. Zhang and Knowles [12] reported that the minimum creep rate of a nickel-based C263 superalloy at 800 °C can be slightly enhanced with increasing amounts of room-temperature pre-tension. Dyson [13,14] examined the creep response of Nimonic 80A at 750 °C after room-temperature pre-tension. The results showed that the material suffered progressive losses of creep strength, life, and fracture ductility as pre-strain increased. Concerning the microstructure, pre-strain increased the density of mobile dislocations on the one hand, resulting in an increase in initial creep rate. On the other hand, dislocations induced by pre-strain will increase the resistance to dislocation movement, which may enhance creep resistance [7]. In addition, it is possible to increase dislocation annihilation rates, which may lead to decreases in creep strain [7]. The density of voids in the grain boundary region increases after pre-strain, which has adverse effects on creep life [7,13,15]. Therefore, the influence of pre-strain on the creep behaviour varies with different alloys. This is related to the temperature and amount of pre-strain, the microstructure, creep testing temperature, stress, and the corresponding creep mechanism. It remains necessary to study the pre-strain effects on creep behaviour for the FGH96 alloy under specific creep test conditions.

Most existing creep models that consider the pre-strain effect are phenomenological [16,17,18,19]. For example, the Norton law for creep was modified to relate the model parameters with pre-strain and temperature [17]. A damage tensor controlled by pre-strain was introduced into the creep damage tensor to reflect the accelerating effect of pre-strain on void nucleation at grain boundaries [18]. Pre-strain effect was involved in internal stress to affect the creep behaviours [19]. Creep mechanisms have not explicitly been included in these models; therefore, feasible creep models considering both the creep mechanism and pre-strain effect need to be established to quantify the creep response changes caused by prior plastic deformation. 

In this study, pre-tension of the FGH96 superalloy was carried out at room temperature, and creep tests were subsequently conducted. The influence of pre-strain on the steady-state creep rates and the influencing mechanism are discussed. Based on the micro-twinning mechanism, a calculation model for the steady-state creep rate was established with the pre-strain effect involved. The numerical results were compared with experimental data to verify the effectiveness of the model. 

## 2. Materials and Methods

The chemical composition (in wt. %) of the FGH96 alloy was: 12.9% Co, 15.7% Cr, 4% Mo, 4% W, 2.1% Al, 3.7% Ti, 0.7% Nb, 0.05% C, 0.03% B, 0.05% Zr, and balanced Ni. The FGH96 alloys were prepared using the powder metallurgy method. The powders were consolidated by hot isostatic pressing (HIP) (1100–1200 °C,120–140 MPa, 3–5 h). Then, the HIPed ingot was hot-extruded and isothermally forged at subsolvus temperatures. After forging, the pancake was heat-treated at 1160 °C for 4 h to obtain a solid solution, and then aged at 760 ℃ for 16 h followed by air cooling.

Uniaxial creep test specimens were subsequently cut from the aged ingot along tangential direction and machined to the specimen dimension (Figure 1). Then, the creep test specimen was pre-strained at room temperature using an INSTRON tensile machine at a crosshead speed of 0.25 mm/min. An extensometer was used to measure the strain during loading and unloading of the tension process. The measured plastic strains after pre-tension were 0.33%, 0.34% 0.35%, 0.43%, 0.47%, 0.48%, 0.60%, 0.61%, 0.63%, 1.01%, 1.05%, 3.01%, and 6.04%, respectively. After pre-tension, the creep tests were carried out at 700 °C and 690 MPa with a duration time of 70 h. 

The metallographic structure was observed by optical microscopy (OM) and the distribution of γ′ precipitates was examined by scanning electron microscopy (SEM: LEO Gemini 1525) before and after pre-tension. The SEM samples were ground, polished, and electro-etched at 3.5 V, 2 A for one second at room temperature. Transmission electron microscopy (TEM) observation foils were sliced from the standard gauge area of the creep test specimen sectioned 90° to the stress axis. The foils were prepared by twin-jet thinning in a solution of 10 vol% HClO4 + 90 vol% C2H5OH at −20 °C. The microstructure of pre-strained and 70 h crept specimens was examined using a JEOL JEM 2100F electron microscope with an operating voltage of 200 kV.

## 3. Creep Model with Pre-Strain Effect

### 3.1. Micro-Twinning-Based Creep Model

Previous studies [1,2,3,4,20,21] on FGH96 superalloys indicated that slip-induced micro-twinning was the dominate creep mechanism at creep temperatures of 650–750 °C and stress of 690–810 MPa. Therefore, the micro-twinning mechanism was considered in the creep model.

According to the micro-twinning mechanism [22], after the movements of two a/6<112> partial dislocations, twins were induced in the γ matrix and pseudo-twins (PTs) were induced in the secondary γ′ matrix. The PTs exhibited an orthorhombic structure which involved Al–Al nearest neighbours, are differently from those remaining in the secondary γ′ with the L1_2_ structure. These PTs have high energies and impede the movements of a/6<112> dislocations. Diffusion-mediated reordering near the {111} planes yielded twinned parts with the ideal L1_2_ structure. The rearranged atomic lattice formed a mirror-image relationship with the original lattice before the PT formation, which meant that true twins (TT) were formed. The transformation from high-energy PTs to low-energy TTs made the a/6<112> dislocations slip again, leading to continuous creep deformation. Therefore, the formation of PTs was induced by the shear of dislocations, while the formation of TTs is thermally mediated.

As shown in Figure 2, in order to establish the theoretical creep model, it was assumed that the secondary γ′ had a cubic morphology and was uniformly distributed in the γ matrix; the tertiary γ′ was ignored due to its much smaller size and lower volume fraction. The energy for the formation of twins in the matrix in a unit cell is expressed as:(1)$$G_{\mathrm{tm}}=\Gamma_{\mathrm{tm}}\left(L^{2}-d_{2}^{2}\right)$$
where $\Gamma_{\mathrm{tm}}$ is the formation energy of two-layer true twins in matrix, $d_{2}$ is the cubic length of the secondary γ′, and *L* is the length of the unit cell.

The energy for the formation of twins in the secondary γ′ consisted of two parts. One was the energy for the formation of true twins, written as
(2)$$G_{\mathrm{tt}}=\Gamma_{\mathrm{tt}}d_{2}^{2}$$
where $\Gamma_{\mathrm{tt}}$ is the formation energy of the two-layer true twins. The other one is the energy required for reordering to accomplish the transformation from PTs to TTs, $G_{\mathrm{ord}}$, which can be related to the time needed for the formation of true twins, expressed as [23,24]:(3)$$G_{\mathrm{ord}}=\Delta \Gamma \exp\left(-Kt_{\mathrm{twin}}\right)d_{2}^{2}$$
where $\Delta \Gamma =\Gamma_{\mathrm{pt}}-\Gamma_{\mathrm{tt}}$, $\Gamma_{\mathrm{pt}}$ is the formation energy for the two-layer PTs. *K* denotes the reordering rate expressed as $K=D_{\mathrm{ord}}/x^{2}$, with $D_{\mathrm{ord}}$ being the diffusion coefficient for the reordering and *x* being the short range diffusion length, usually obtained as $x=2b_{\mathrm{tp}}$, where $b_{\mathrm{tp}}$ is the magnitude of the Burgers vector [23,24].

Similarly to the model developed by Karthikeyan et al. [25], considering the movements of a/6<112> dislocations in two-layer {111} planes, the energy balance at the steady-state stage can be expressed as:(4)$$2\left|\tau_{}^{\alpha }\right|b_{\mathrm{tp}}L^{2}=G_{\mathrm{ord}}+G_{\mathrm{tt}}+G_{\mathrm{tm}}$$
where $\tau_{}^{\alpha }$ is the effective resolved shear stress in a *α* slip system. The above equation indicates that the work performed by external force is used for the formation of true twins in matrix and secondary γ′, and the energy for the reordering. The energy for reordering should be positive; therefore, a requirement for the stress is written as $2\left|\tau_{}^{\alpha }\right|b_{\mathrm{tp}}L^{2}>\Gamma_{\mathrm{tt}}d_{2}^{2}+\Gamma_{\mathrm{tm}}\left(L^{2}-d_{2}^{2}\right)$, which is also equivalent to $2\left|\tau_{}^{\alpha }\right|b_{\mathrm{tp}}>\Gamma_{\mathrm{tt}}f^{2}+\Gamma_{\mathrm{tm}}\left(1-f^{2}\right)$ with $f=d_{2}^{}/L$. 

Combining Equations (1)–(4) yields:(5)$$t_{\mathrm{twin}}=\frac{1}{K}\ln\frac{f^{2}\Delta \Gamma }{2b_{\mathrm{tp}}\left|\tau_{}^{\alpha }\right|-f^{2}\Gamma_{\mathrm{tt}}-\Gamma_{\mathrm{tm}}+f^{2}\Gamma_{\mathrm{tm}}}$$

Using the Orowan equation [26], the shear rate is expressed as [25]
(6)$${\dot{\gamma }}^{\alpha }=\rho_{tp}^{\alpha }b_{tp}^{}v_{}^{\alpha }=\rho_{tp}^{\alpha }b_{tp}^{}\frac{b_{\mathrm{tp}}}{t_{twin}}$$
where $\rho_{tp}^{\alpha }$ is the mobile dislocation density, and $v_{}^{\alpha }$ is the average dislocation velocity which is obtained considering the time for the formation of true twins in secondary *γ′* and ignoring the time for dislocation slip.

Combining Equations (5) and (6), the shear rate is obtained as:(7)$${\dot{\gamma }}^{\alpha }=\frac{\rho_{tp}^{\alpha }b_{tp}^{2}D_{\mathrm{ord}}/x^{2}}{\ln\left[f^{2}\Delta \Gamma /\left(2b_{\mathrm{tp}}\left|\tau_{}^{\alpha }\right|-f^{2}\Gamma_{\mathrm{tt}}-\Gamma_{\mathrm{tm}}+f^{2}\Gamma_{\mathrm{tm}}\right)\right]}$$

If the dislocation density is assumed to be unchanged in the steady-state creep stage, the shear rate can be expressed as:(8)$${\dot{\gamma }}^{\alpha }=\frac{{\dot{\gamma }}_{0}^{\alpha }}{\ln\left[f^{2}\Delta \Gamma /\left(2b_{\mathrm{tp}}\left|\tau_{}^{\alpha }\right|-f^{2}\Gamma_{\mathrm{tt}}-\Gamma_{\mathrm{tm}}+f^{2}\Gamma_{\mathrm{tm}}\right)\right]}$$
where ${\dot{\gamma }}_{0}^{\alpha }=\rho_{tp}^{0}b_{tp}^{2}D_{\mathrm{ord}}/x^{2}$ is the initial shear rate, which is related to the initial dislocation density ($\rho_{tp}^{0}$).

For uniaxial creep, considering multiple slip, the relationship between uniaxial creep rate and shear rate can be written as [27,28]:(9)$${\dot{\varepsilon }}^{cr}=\frac{1}{M}{\dot{\gamma }}^{\alpha }$$
where *M* is the Taylor factor [29].

In addition, the resolved shear stress can be expressed as [29,30]:(10)$$\tau^{\alpha }=\frac{\sigma }{M}$$
where σ denotes the uniaxial stress. 

Combining Equations (8)–(10), the uniaxial creep rate is obtained as:(11)$${\dot{\varepsilon }}^{cr}=\frac{\rho_{tp}^{0}b_{tp}^{2}D_{\mathrm{ord}}/x^{2}}{M\cdot \ln\left[f^{2}\Delta \Gamma /\left(2b_{\mathrm{tp}}\frac{\sigma }{M}-f^{2}\Gamma_{\mathrm{tt}}-\Gamma_{\mathrm{tm}}+f^{2}\Gamma_{\mathrm{tm}}\right)\right]}$$

The above equation relates the steady-state creep rate to the uniaxial stress. The strain rate is proportional to the initial density a/6<112> partial dislocations and average diameter of secondary γ′.

### 3.2. Creep Model with Pre-Strain Effect

In order to capture the increase in mobile dislocation density with the increase in pre-strain, the Kocks–Mecking [31,32] relationship was adopted to describe the evolution of dislocation densities with time, considering the working hardening and dynamic recovery mechanisms, expressed as [33]:(12)$$\frac{\partial \rho }{\partial \varepsilon }=M(k_{1}\sqrt{\rho }-k_{2}\rho )$$
where ρ is the dislocation density, ε is the applied strain, and *M* is the Taylor factor. *k*_1_ is a temperature-independent parameter reflecting the dislocation accumulation, and *k*_2_ is a temperature-dependent parameter describing the dislocation annihilation [34]. 

Assuming that the dislocation density can be uniquely determined by pre-strain in the pre-tension process, the relationship between dislocation density and pre-strain can be obtained by integrating Equation (12), and is written as:(13)$$\frac{\rho }{\rho_{0}}={\left\{1+\left(\frac{k_{1}}{k_{2}}\rho_{0}^{-1/2}-1\right)\left[1-\exp\left(-\frac{Mk_{2}}{2}\varepsilon \right)\right]\right\}}^{2}$$

Combining Equations (11) and (13), the steady-state creep rate model with the pre-strain effect could be given as:(14)$$\begin{array}{l} {\dot{\varepsilon }}^{cr}=\frac{\rho_{tp}^{0}b_{tp}^{2}D_{\mathrm{ord}}/x^{2}}{M\cdot \ln\left[f^{2}\Delta \Gamma /\left(2b_{\mathrm{tp}}\frac{\sigma }{M}-f^{2}\Gamma_{\mathrm{tt}}-\Gamma_{\mathrm{tm}}+f^{2}\Gamma_{\mathrm{tm}}\right)\right]}\\ \frac{\rho_{tp}^{0}}{\rho_{tp-pre}^{0}}={\left\{1+\left(\frac{k_{1}}{k_{2}}{\left(\rho_{tp-pre}^{0}\right)}_{}^{-1/2}-1\right)\left[1-\exp\left(-\frac{Mk_{2}}{2}\varepsilon_{pn}\right)\right]\right\}}^{2} \end{array}$$
where $\varepsilon_{pn}$ denotes the pre-strain, and $\rho_{tp-pre}^{0}$ is the mobile dislocation density without pre-strain.

## 4. Result and Discussion

### 4.1. Influence of Pre-Strain on the Creep Behaviour

Figure 3a shows the creep strain curve of specimens with different amount of pre-strain. It can be seen that the 70 h creep strain continuously increased with the increase in pre-strain. For specimens with the same amount of pre-strain, the 70 h creep strain exhibited certain dispersity, which might be caused by the variations in the microstructure and internal stress of the specimen. The amount of creep strain at the primary stage and the duration of primary creep stage increased with the increase in pre-strain, while the steady-state stage still dominated. Compared with that of the unstrained specimen, the creep strain of pre-strained specimens accumulated rapidly in the first few hours, and then crept at a higher steady-state rate. For the specimen with 6.04% pre-strain, the creep strain exceeded 0.2% in the first 5 h, far more than the creep strain after 70 h of specimens that had pre-strain less than 0.6%. 

As shown in Figure 3b, the steady-state creep rate increased with the increasing amount of pre-strain. Even a small amount of pre-strain had a clear impact on the steady-state creep rate, indicating that the creep rate of FGH96 alloy at 700 °C/690 MPa was sensitive to room-temperature pre-strain. For the specimen with 3% pre-strain, the steady-state creep rate was more than 10 times greater than that of unstrained specimen, which means that a large amount of pre-strain will result in a significant loss of creep resistance. 

### 4.2. Influence of Pre-Strain on Microstructure

The grain morphology and distribution of γ′ precipitates before and after pre-tension are shown in Figure 4. The unstrained and 6.04% pre-strained specimens had a similar grain morphology with an average grain size of approximately 40 μm. The morphology and distribution of both secondary γ′ and tertiary γ′ showed no obvious differences. Most of the secondary γ′ particles were spherical or cuboid in shape, with concave faces, similarly to the results of previous studies [4,35]. For unstrained and 6.04% pre-strained specimens, the volume fractions of secondary γ′ of were 39% and 41%, and the average diameters were 162 nm and 148 nm, respectively. This indicated that room-temperature pre-tension within 6.04% plastic strain had no obvious influence on the morphology and distribution of γ′ precipitates. 

Figure 5 shows the dislocation distribution in the FGH96 alloy after various degrees of room-temperature pre-strain. After 0.33% pre-strain, independent dislocations appeared, and the dislocations were mainly located in the matrix (Figure 5b). After 1.01% pre-strain, a large number of dislocations were found both in the matrix and the γ′/γ interface (Figure 5c). The dislocation density was much higher than that of the 0.33% pre-strained specimen. When the pre-strain reached 6.04%, continuous dislocation lines appeared and the dislocation lines were parallelly distributed (Figure 5d). At this point, significant plastic deformation and strain strengthening occurred. The dislocation density inside the alloy significantly increased. 

The creep deformation of the Ni-based superalloy involved two parts: dislocation slipping in the matrix channel and dislocation cutting through γ′ particles. In the primary stage of creep, creep deformation was mainly provided by dislocation slip in the matrix channel. The change in dislocation density under different amounts of pre-strain will affect the creep behaviour of FGH96 alloys to varying degree. For the 0.3% pre-strained specimen, pre-strain-induced dislocations slipped in the matrix channel at the beginning of creep. Compared with the unstrained specimen, the existence of pre-strain-induced dislocations reduced the time for dislocation formation and accelerated the progression of dislocation slip in the primary creep stage. For specimens with 1.01% and 6.04% pre-strain, the mobile dislocation densities increased significantly, which further accelerated dislocation movement and increased the strain rate during primary creep stage. In addition, the release of misfit stress, caused by the formation of interface dislocations during room-temperature pre-deformation, could reduce the hindrance to dislocation movement to some extent. This might also be a factor contributing to the increase in creep rate.

According to the Orowan equation [26], two competing effects can be introduced by pre-strain in terms of the creep strain rate. On the one hand, the dislocation density induced by room-temperature pre-strain impedes dislocation movement, thereby reducing the average moving velocity of dislocations and resulting in a decrease in creep rate, namely, the creep weakening effect. On the other hand, the dislocations induced by pre-strain also remain mobile, thus leading to an increase in creep rate, namely, the creep enhancement effect. The effect of pre-deformation on the creep rate mainly depends on which of these two competing factors dominates. It can be seen from the experimental results that creep enhancement effect dominated. The increase in mobile dislocation density caused by pre-strain was the main reason for the increase in primary creep rate of the FGH96 alloy.

Figure 6 shows the microstructure of the FGH96 alloy after 70 h creep. Compared with the microstructure before creep, the dislocation density after 70 h creep was much higher. For the 0.34% pre-strained specimen, the dislocation density was significantly increased during creep and a large number of dislocations were entangled at the γ′ and γ interface (Figure 6a). For the 1.0% pre-strained specimen, continuously distributed stacking faults were found after 70 h creep (Figure 6b). This was a result of the interaction between dislocations and γ′ particles. It was obvious that greater creep deformation occurred for the 1.0% pre-strained specimen. The numerous dislocations induced by pre-strain accelerated the creep. 

The influence of pre-strain on the creep rate depends on the creep mechanisms. The creep mechanism mainly included three types: dislocation gliding along the slip plane induced by thermal activation at a high stress level, dislocation movement induced by vacancy diffusion at a medium stress level, and diffusion creep controlled by atom diffusion at a low stress level. The creep mechanism was dependent on the testing temperature, stress, amount of pre-strain, and the resultant microstructure. According to existing experiments on FGH96 superalloys in the literature [1,2,3,4,20,21], slip-induced micro-twinning is the dominate deformation mechanism at 700 °C/690 MPa. An increase in pre-strain increases the dislocation density, thus resulting in a higher steady-state creep rate. 

### 4.3. Validation of the Creep Rate Model

The parameters used in the model can be categorised as material constants, microstructure information, and fitted parameters. The material constants are listed in Table 1, including the magnitude of the Burgers vector (*b*_tp_), the formation energy of two-layer PTs in secondary *γ′* (Γ_pt_), the formation energy of two-layer TTs in secondary *γ′* (Γ_tt_), the formation energy of two-layer TTs in the *γ* matrix (Γ_tm_), the diffusion coefficient for reordering (*D*_ord_), and the diffusion length (*x*). Reordering involves the diffusion of Ni and Al atoms; a definite value of *D*_ord_ was still unavailable. The diffusion coefficient of Al atoms in Ni_3_Al was obtained as [36]:(15)$$D_{Al}^{*}=5.05\times 10^{-7}\exp(-\frac{243\;\mathrm{KJ}/\mathrm{mol}}{RT})\;m^{2}/s$$
where *R* is the ideal gas constant (8.314 $J\cdot {\mathrm{mol}}^{-1}\cdot K^{-1}$) and T is the thermodynamic temperature. 

The self-diffusion coefficient of Ni atoms in Ni_3_Al was obtained as [37]:(16)$$D_{Ni}^{*}=3.59\times 10^{-4}\exp(-\frac{303\;\mathrm{KJ}/\mathrm{mol}}{RT})\;m^{2}/s$$

In the simulation, *D*_ord_ was set as the average value of the two diffusion coefficients [25], i.e., $D_{ord}=\frac{D_{Al}^{*}+D_{Ni}^{*}}{2}$. The diffusion length, *x*, was taken as twice the value of *b*_tp_ [25].

The volume fraction and diameter of the secondary *γ′* were obtained from the experiments. As shown in Section 2, pre-strain within 6% did not have obvious effects on *γ′* precipitations; therefore, the volume fraction and diameter of γ′ in the sample without pre-strain were adopted, which means that the volume fraction, ϕ, was 40%, and the average diameter, $d_{2}$, was 162 nm. The parameter f was obtained according to the volume fraction, expressed as $f=\phi^{1/3}$. The Taylor factor, *M*, was 3.06; this is commonly adopted in FCC materials [30]. The values of $\rho_{tp-pre}^{0}$, *k*_1_, and *k*_2_ were calibrated by experimental data; their values are listed in Table 2. These three parameters were determined as follows. According to Equation (11), $\rho_{tp-pre}^{0}$ was determined using the steady-state strain rates of the sample without pre-strain. Then, the values of *k*_1_ and *k*_2_ were calibrated according to Equation (13) using the steady-state strain rates of the pre-strained samples. Finally, Equation (14) was adopted to calculate the creep rates using the parameters shown in Table 1 and Table 2. 

**Table 1 materials-16-03874-t001:** Parameters of the material constants.

*b*_tp_ (Å)	Γ_pt_ (J/m^2^)	Γ_tt_ (J/m^2^)	Γ_tm_ (J/m^2^)	*D*_ord_ at 973 K (m^2^/s)	*x*
1.44 [38]	0.7 [25]	0.02 [25]	0.03 [39]	3.24 × 10^−20^	2 *b*_tp_ [25]

As shown in Figure 7, the predicted steady-state creep rates increased with the increase in pre-strains and finally reached saturation value, which was related to the evolution of dislocation densities because the creep rate was proportional to dislocation density. According to Equation (14), the saturation dislocation density can be obtained as ${\left(\frac{k_{1}}{k_{2}}\right)}^{2}$; it was equal to 1.75 × 10^13^ m^−2^. Pre-strains of 6% had no obvious influence on the morphology and distribution of γ′ precipitates and pre-strains mainly affected the density of a/6<112> dislocations; therefore, an exponential function of dislocation density was adopted to identify the pre-strain effect.

The comparisons between numerical calculated creep rates and experimental data are shown in Figure 8. The steady-state creep rates increased with the increase in pre-strain, and predicted data showed good agreement with experiment data, which indicated that the creep model proposed in this study could capture the pre-strain effect, considering the micro-twinning mechanism. The difference between the predicted results and the experimental data may result from the variations in microstructure, such as the volume fraction and the average diameter of γ′ precipitates. The pre-strain effects were included into the creep models by extrapolating the damage variable, internal stress, or strain rates from the literature [7,17,18,19]. Although creep resistance or creep enhancement effects induced by pre-strain could be captured in these models, they did not explicitly consider the creep mechanisms. In this study, the increase in dislocation density induced by pre-strain was reflected, and the steady-state creep rates were obtained by explicitly considering the slip-induced micro-twinning mechanism. The proposed model showed good applicability in micro-twinning-induced creep deformation, and provided an effective means for predicting the steady-state creep rates of materials with pre-strain.

## 5. Conclusions

Pre-tension experiments with various plastic strains at room temperature and subsequent creep tests at 700 °C/690 MPa were conducted for FGH96 alloys, and the microstructures after room-temperature pre-stain and high-temperature creep were analysed. Based on the micro-twinning mechanism, a steady-state creep rate model was established; the density of mobile dislocations evolved with pre-strain. The calculated steady-state creep rates were compared with the experimental data, and good agreement could be obtained. The main conclusions can be drawn as follows:
(1)The steady-state creep rate and 70 h creep strain continuously increased with the increase in pre-strains. Compared with an unstrained specimen, the creep strain of pre-strained specimens accumulated rapidly in the first few hours, followed by a higher steady-state creep rate. For specimens with more than 1% pre-strain, the steady-state creep rate was more than 10 times greater than that of the unstrained specimen.(2)Room-temperature pre-tension within 6.04% plastic strain had no obvious influence on the morphology and distribution of γ′ precipitates, whereas the dislocation density continuously increased with the increase in pre-strains. The increases in mobile dislocation density after pre-strain were the main reasons for the increase in creep rates of FGH96 alloys.(3)The predicted steady-state creep rates showed good agreement with the experimental data; the creep model proposed in this study could capture the pre-strain effect while considering the micro-twinning mechanism.

## Figures and Tables

**Figure 1 materials-16-03874-f001:**
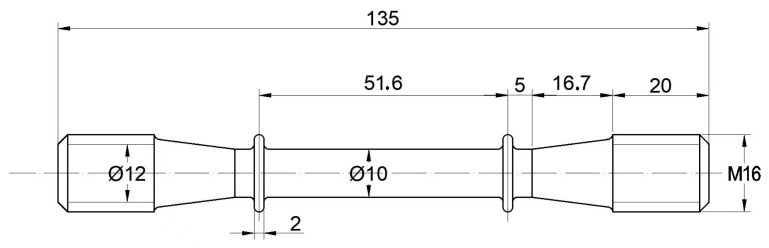
Dimensional drawing of the creep test specimen (unit: mm).

**Figure 2 materials-16-03874-f002:**
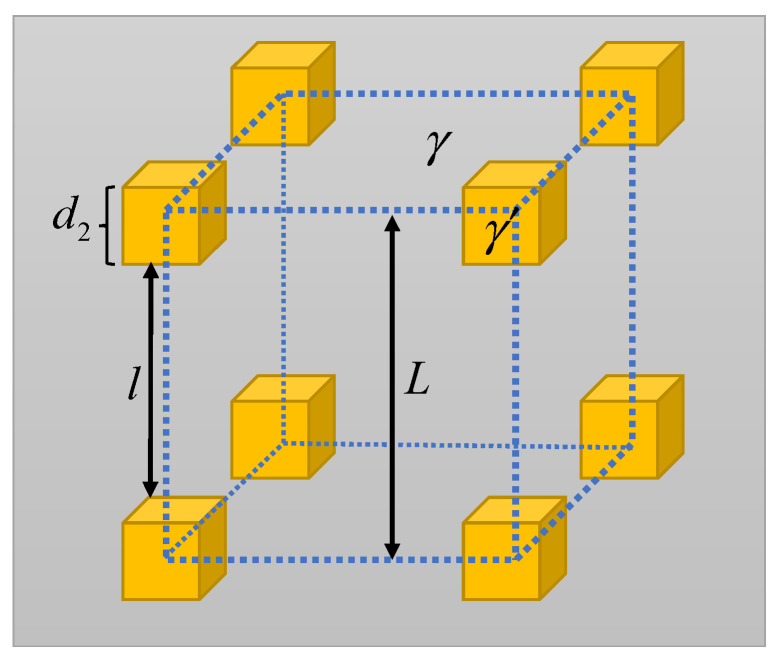
Illustrations of the γ matrix and secondary γ′.

**Figure 3 materials-16-03874-f003:**
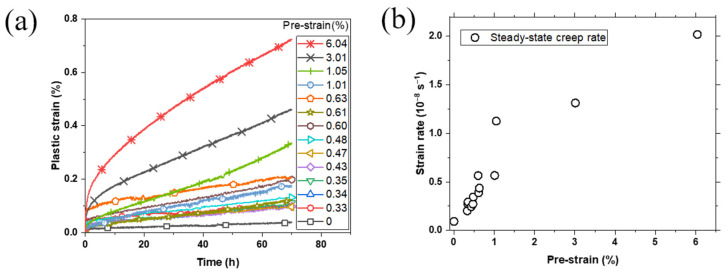
(**a**) Creep curves and (**b**) steady-state creep rates at 700 °C/690 MPa.

**Figure 4 materials-16-03874-f004:**
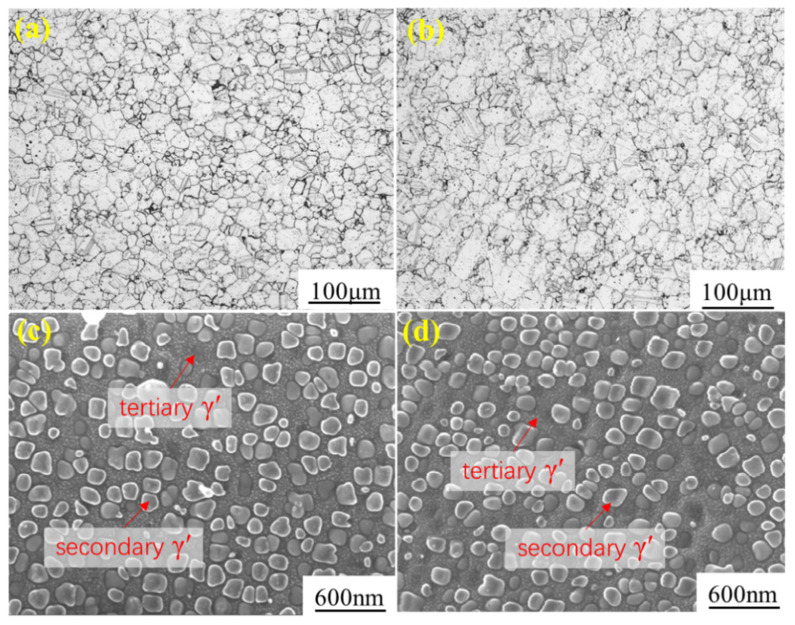
The grain morphology (**a**,**b**) and γ′ distribution (**c**,**d**) of FGH96 before and after pre-strains: (**a**) OM unstrained, (**b**) OM 6.04% pre-strained, (**c**) SEM unstrained, and (**d**) SEM 6.04% pre-strained.

**Figure 5 materials-16-03874-f005:**
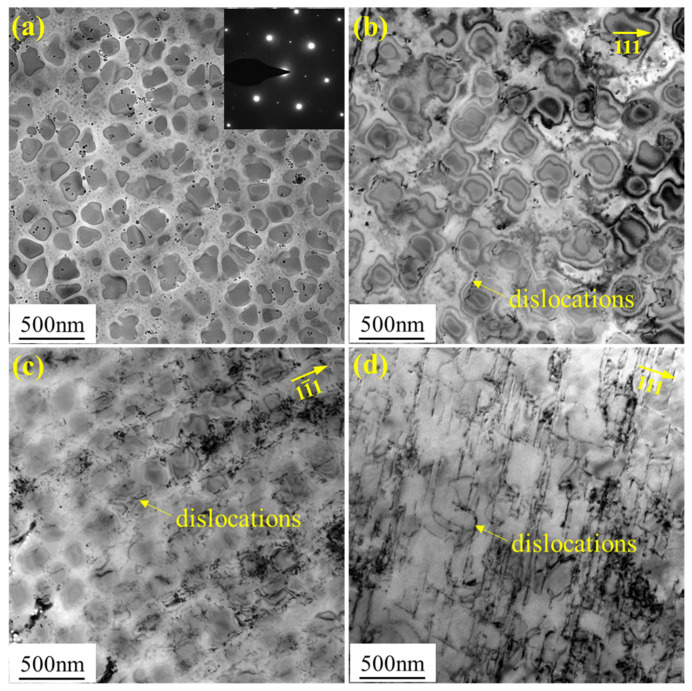
TEM analysis of the FGH96 alloy after various degrees of pre-strains: (**a**) un-strained, (**b**) 0.33% pre-strain, (**c**) 1.01% pre-strain, and (**d**) 6.04% pre-strain.

**Figure 6 materials-16-03874-f006:**
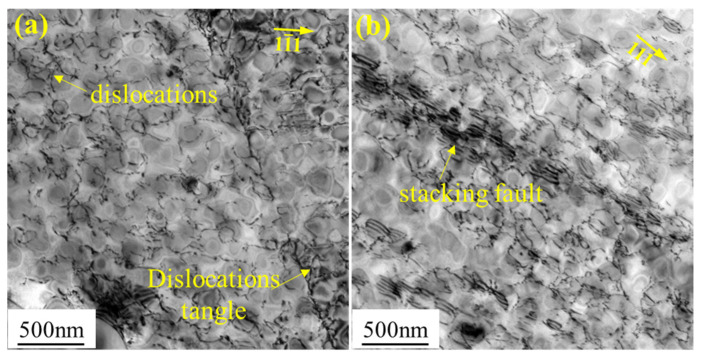
TEM analysis of the FGH96 alloy after 70 h creep: (**a**) 0.34% pre-strained sample, (**b**) 1.05% pre-strained sample.

**Figure 7 materials-16-03874-f007:**
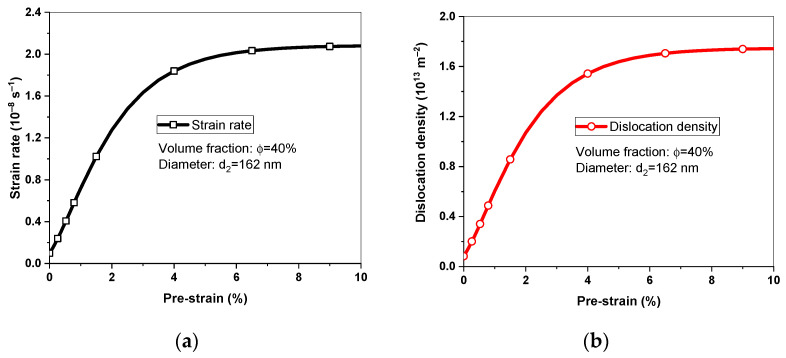
Predicted (**a**) steady-state creep rates and (**b**) dislocation density under different pre-strains.

**Figure 8 materials-16-03874-f008:**
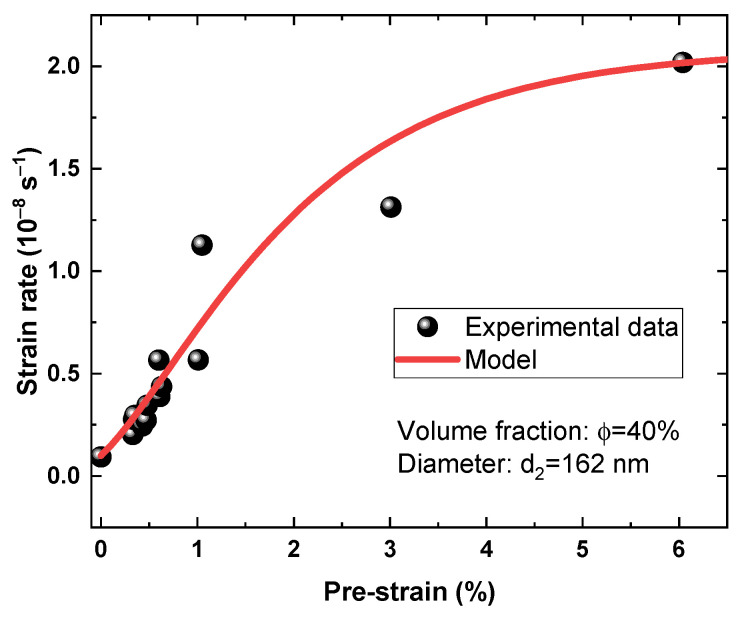
Comparisons between the numerical steady-state creep rates and the experimental data.

**Table 2 materials-16-03874-t002:** Fitted parameters.

$\rho_{tp-pre}^{0}$ (m^−2^)	*k*_1_ (m^−1^)	*k* _2_
8.27 × 10^11^	1.7490 × 10^8^	41.83

## Data Availability

The data presented in this study are available on request from the corresponding author.

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
