# Peer review of "Experiment and Modelling of the Pre-Strain Effect on the Creep Behaviour of P/M Ni-Based Superalloy FGH96"

_materials, 2023, doi:10.3390/ma16103874_

Round 1
Reviewer 1 Report
This paper is an interesting study investigating the effect of introducing prestrain in nickel-based alloys on creep behavior. However, due to some errors and insufficient discussion, the reviewer does not recommend publication of this paper in Materials without appropriate corrections.
The points that need to be corrected are as follows.
1. The links to Figure and Table are not showing up correctly.
2. Does equation (14) include the density of moving dislocations at steady-state creep? This value is required to obtain the steady-state strain rate based on the viscous glide model.
3. Line 243 says "As the γ' phase was dispersely distributed, the slip resistance of dislocations during creep was overall reduced, thereby increasing the initial creep strain.".
Does this mean that the distribution of γ' has changed due to the pre-deformation? If so, this contradicts the statement in line 218.
4. Are the diffraction conditions of the TEM images appropriate for visualizing dislocations and misfit strain fields? The reviewer recommends that the authors describe the TEM observation conditions to demonstrate the validity of the data.
5. There are several problems with the paragraph starting at line 235. The elastic field at the γ/γ' interface should not be considered a direct creep resistance, but a factor affecting the growth behavior of the γ' particles. No clear elastic strain field seems to be visualized in Fig. 4. The technical term for dislocations placed at the interface between the matrix and the semi-coherent particles is "misfit dislocations".
6. The authors conclude that pre-straining increases the number of mobile dislocations. If so, the statement from line 283 is incorrect. If the creep rate is controlled by the viscous slip of the dislocations, an increase in mobile dislocation density increases the strain rate. That is, it causes creep weakening.
7. The value listed in Table 2 as the dislocation density without pre-strain appears to be too small. The authors state that this value was calibrated from experimental data, but they should explain in the paper how they did it and how they obtained the dislocation density data they used. The saturated dislocation density estimated from equation (14) on line 329 is also too low and should be discussed in comparison to the steady state dislocation density estimated from TEM observations.
Author Response
Dear Reviewer
We appreciate your comments on our manuscript. These comments are very helpful for improving our manuscript. We have considered these comments carefully and made corrections. we do provide a specific response (bold) to each comment/suggestion of the reviewer (blue italicized). All important changes introduced in the revised manuscript are highlighted for easy identification (yellow highlight). Please find the attached file for details.

Reviewer 2 Report
Dear Authors,
Please find comments on the paper "Experiment and modelling of pre-strain effect on creep behaviour of powder metallurgy nickel-based superalloy". The paper describes the creep tests performed on a Nickel-based superalloy produced by powder metallurgy. It proposes a steady-state creep rate model considering the micro-twinning mechanisms and pre-strain effects. The research performed in this work fits the scope of Materials, and it would interest the readers. The manuscript presents good-quality results in the field. However, it needs some English and text editing for paper improvement and a better understanding; please consider the following comments,
Revised the English spelling of the title and change it.
The information in the results and discussion section should be more organized since it is difficult to understand, particularly the description of creep results in 4.1 and the Influence of pre-strain on microstructure and creep behaviour in 4.2.
Use appropriate figures and tables referencing and equation numbering format.
Compare the proposed model's results with those from other models in the discussion.
The authors must address the English and text edition of the manuscript before it can be considered for publication in Materials.
English needs to be revised to improve the structure of the sentences. Sometimes it takes work to understand it. Some text editing is also needed since the manuscript presents flaws and punctuation errors.
Author Response

(The authors gave the same response as above.)

Reviewer 3 Report
In this work, the authors have reported the effect of pre-strain on the creep deformation behavior of PM FGH96 alloy and proposed a steady-state creep rate model. However, although the work is interesting, the manuscript needs extensive revision before publication. The following points are detracting:
- The title “powder metallurgy nickel-based superalloy” should be changed.
- How pertinent is it to study the creep effect of aero-engine disc material? The authors should highlight the significance of their work. What is the importance of pre-strain?
- What is the basis for choosing the given parameters?
- The dimensional drawing of the test specimen must be added. The experimental section should include a detailed explanation of sample preparation and TEM characterization.
- Zone axis and g vectors must be added to the TEM micrographs (Figs. 4-5).
- Detailed TEM analysis is required. The authors need to explain the dislocation-based mechanisms better. Strong statements have been written without any supportive evidence.
- A thorough discussion is required to propose the exact dislocation mechanism.
- How was dislocation density measured?
- Fig. 7 numerical model and experimental data seem to match only at lower pre-strain.
- An extensive English revision is required.
- An extensive English revision is required.
Author Response

(The authors gave the same response as above.)

Round 2
Reviewer 1 Report
This manuscript has been appropriately revised.
As a reviewer, I propose to accept this manuscript.
Reviewer 3 Report
The manuscript can be accepted in the present from.
Seems to be ok.